

# Microbial activity responses to water stress in agricultural soils from simple and complex crop rotations

Jörg Schnecker[1], D. Boone Meeden[2], Francisco Calderon[3], Michel Cavigelli[4], R. Michael Lehman[5], Lisa K. Tiemann[6], and A. Stuart Grandy[2]

[1]Department of Microbiology and Ecosystem Science, University of Vienna, Vienna, 1090, Austria
[2]Department of Natural Resources and the Environment, University of New Hampshire, Durham, NH 03824, USA;
[3]College of Agricultural Sciences, Oregon State University, Corvallis, OR 97333, USA;
[4]Sustainable Agricultural Systems Laboratory, USDA-ARS, Beltsville, MD 20705, USA;
[5]North Central Agricultural Research Laboratory, USDA-ARS, Brookings, SD 57006, USA;
[6]Department of Plant, Soil and Microbial Science, Michigan State University, East Lansing, MI 48824, USA

*Correspondence to*: Jörg Schnecker (joerg.schnecker@univie.ac.at)

**Abstract.** Increasing climatic pressures such as drought and flooding challenge agricultural systems and their management
globally. How agricultural soils respond to soil water extremes will influence biogeochemical cycles of carbon and nitrogen in these systems. We investigated the response of soils from long term agricultural field sites under varying crop rotational complexity to either drought or flooding stress. Focusing on these contrasting stressors separately, we investigated soil heterotrophic respiration during single and repeated stress cycles in soils from four different sites along a precipitation gradient (Colorado, MAP 421 mm; South Dakota, MAP 580 mm; Michigan, MAP 893 mm; Maryland, MAP 1192 mm); each site had
two crop rotational complexity treatments. At the driest (Colorado) and wettest of these sites (Maryland) we also analyzed microbial biomass, six potential enzyme activities and $N_2O$ production, during and after individual and repeated stress cycles. In general, we found site specific responses to soil water extremes, irrespective of crop rotational complexity and precipitation history. Drought usually caused more severe changes in respiration rates and potential enzyme activities than flooding. All soils returned to control levels for most measured parameters as soon as soils returned to control water levels following drought
or flood stress, suggesting that the investigated soils were highly resilient to the applied stresses. The lack of sustained responses following the removal of the stressors may be because they are well in the range of natural in situ soil water fluctuations at the investigated sites. Without inclusion of plants in our experiment, we found that irrespective of crop rotation complexity, soil and microbial properties in the investigated agricultural soils were more resistant to flooding but highly resilient to drought and flooding, during single or repeated stress pulses.

**1 Introduction**

Future climate scenarios predict increasingly frequent and extreme weather events, with both more severe droughts and flooding (Stocker et al., 2013). How these shifts in precipitation patterns affect agricultural systems is of special interest due



to their roles in food security and global carbon and nutrient cycling, both of which are likely to alter with climate change (Bowles et al., 2018).

Soil microorganisms, which drive nutrient and carbon cycling, will regulate how soils respond to these shifts in precipitation patterns. Both drought and flooding influence microbial processes and functions (Schimel, 2018), which in turn may feedback to plant-soil interactions (Canarini and Dijkstra, 2015; Kaisermann et al., 2017). For example, reduced water content in soils can cause microbial death or sporulation and thereby strongly reduce overall microbial activity (Herron et al., 2009). Even under less severe reductions in soil water content, microbial activity decreases since diffusion, microorganisms' main means

of substrate transport (Bailey et al., 2017; Schimel, 2018; Tecon and Or, 2017), is reduced in concert with reduced connectivity of microorganisms and soil organic matter and nutrients (Linn and Doran, 1984; Schnecker et al., 2019). Lower soil water may also lead to higher soil solute concentrations, enhancing osmotic stress for microorganisms (Killham and Firestone, 1984; Wood, 2015).

As soils recover from drought and are re-wetted, numerous studies have observed an increase of respiration rates that often

exceed control levels for days after rewetting (Birch, 1958; Fierer and Schimel, 2002; Li et al., 2010). This 'Birch effect' (Birch, 1958) is associated with an increase of available dissolved organic C (DOC) through microbial death during drought (Schimel, 2018) or caused by the lower drought-sensitivity of extracellular enzymes compared to microorganisms, which results in enzymes solubilizing SOM that is not taken up by the inactive microbes until rewetting (Schimel, 2018; Steinweg et al., 2013).


Under water-saturated conditions, soils run the risk of oxygen ($O_2$) deficiency leading to less efficient microbial energy generation and production of potent greenhouse gases (Berglund and Berglund, 2011; Linn and Doran, 1984; Randle-Boggis et al., 2018; Smith et al., 2003). Rewetting can displace $CO_2$ from soil pores, causing a degassing that can in turn affect microbial metabolism (Calderón and Jackson, 2002). In agricultural systems, anaerobic conditions or repeated changes from

wet to dry conditions can lead to of nitrous oxide ($N_2O$) production and alter the cycling of bioavailable N (Bowles et al., 2018; Davidson, 1992; Muhr et al., 2008).

How microorganisms respond to stress determines if specific microbial processes and functions can withstand or be buffered against stress. Resistance to stress is the ability of microbial communities to withstand prevailing stressors (Allison and Martiny, 2008) and reduce the amplitude of the stress response. Along with resistance, stress response can be defined by

resilience, which characterizes the duration of the stress response. A resilient microbial community quickly returns to pre-stress levels (Allison and Martiny, 2008). In addition to these initial or one-time microbial reactions to abiotic stress, microbial communities may also adapt to re-occurring stressors (DeAngelis et al., 2010; Evans and Wallenstein, 2012) by progressively reducing the initial amplitude and/or duration of the stress response with each recurring stress event.

Adaptation to recurring stress is more likely to occur with greater microbial diversity, which often correlates with functional

redundancy (Girvan et al., 2005) and the probability that members of the community have physiological traits that improve their stress responses (Griffiths and Philippot, 2013). Land management history can affect soil microbial diversity, which in

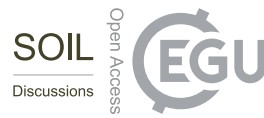

turn can affect how soil responds and recovers from disturbances (Jackson et al., 2003). Microbial diversity in agricultural soils has been shown to increase with crop rotation complexity (Tiemann et al., 2015; Venter et al., 2016) and especially the introduction of cover crops (Vukicevich et al., 2016). These management practices also result in increased soil microbial

biomass (McDaniel et al., 2014) and organic matter (Ding et al., 2006; McDaniel et al., 2014), while reducing agroecosystem N loss and improving crop yields under climate stress (Bowles et al., 2020). While there is evidence to support a link between crop rotation complexity and resilience of crops yields under climate stress such as drought or flooding, it remains unclear if microbial communities in these complex cropping systems are also resilient.

Drought and flooding are contrasting forms of stress for microorganisms and challenge them in very different ways; however,

drought and flooding are usually studied simultaneously. Flooding in particular is usually studied only as rewetting events after drought (Birch, 1958; Schimel, 2018). Therefore, classic drought-rewetting experiments provide only limited insight into microbial response to the individual stressors: drought and flooding.

The aim of this study was to test microbial responses to one-time and recurring episodes of drought or flooding, and whether and how these responses are moderated by a history of crop diversification. In a laboratory incubation we manipulated water

regimes in soils from four long-term crop rotation experiments across the USA. At each site we selected a low (one or two crops, "simple rotation") vs. high (>3 crops, "complex rotation") diversity rotation for comparison. The sites range from low (Colorado) to intermediate (Michigan and South Dakota) and high (Maryland) precipitation. Soils from different regions were chosen to examine whether potential adaptations to drought or flooding depend on historical climate. Replicate sets of soil samples were either alternately dried and rewetted to optimum moisture content ("drought"), alternately flooded and dried to

optimum moisture ("flooding"), or maintained at a constant water content (control). We monitored heterotrophic soil respiration ($CO_2$ production) during five moisture stress cycles. Additionally, we determined microbial biomass, enzyme activities, and N pools and fluxes during the first and last stress cycle in soils from sites with the precipitation extremes.

## 2 Material and Methods

### 2.1  Sampling sites

Soils were collected in October 2015 from long-term crop rotation experiments at USDA-ARS sites in Akron, Colorado (CO), Beltsville, Maryland (MD), and Brookings, South Dakota (SD), and at the W.K. Kellogg Biological Station (KBS) Long-Term Ecological Research Site (LTER), Michigan (MI). All sites maintain field experiments that include treatments with different rotation lengths. Composite topsoil samples from within the first 10 cm were collected from three (Colorado) to four (Maryland, South Dakota, Michigan) field plot replicates in simple (2 crops in rotation) and complex rotational treatments (3-

4 crops in rotation) each.  Soils from Maryland, South Dakota, and Michigan were sampled under corn and those from Colorado were sampled under wheat. Site descriptions can be found in table 1 and in (Cavigelli et al., 2008; Lehman et al., 2017; Tiemann et al., 2015; White et al., 2019).



## 2.2 Experimental setup

After sampling, soils were sieved and shipped on ice to the University of New Hampshire and refrigerated at 5oC for less than
one week. Approximately 30 g soil from each replicate plot were weighed into 100 mL plastic cups resulting in a total of
twenty-seven cups per replicate from Colorado and Maryland and six for South Dakota and Michigan. Soils in the microcosms
were adjusted to 50% water holding capacity (WHC). One set of cups was covered with parafilm and kept at constant water
content by replacing evaporated water once a week and after every $CO_2$ measurement, over the course of 165 days. One set of
cups was subjected to drought and another to flooding (Fig.1). All soil microcosms were kept at a constant temperature of
25°C. Microcosms for the drought treatment were allowed to gradually dry out over the course of 3 days, kept at peak drought
for 4 days, and slowly brought back to 50% WHC by adding one third of the evaporated water every day for three days to
avoid even short time flooding effects. Microcosms for the flooding treatment were gradually brought to 100% WHC but not
higher to avoid submerging the soils in water over the course of three days, were then kept at 100% WHC for 4 days and were
then kept open to dry back to 50% WHC again within 3 days. Drought and wetting were repeated after two weeks of soils
being held at constant WHC. Soils were subjected to a total of five stress cycles during the first 125 days of the total 165-day-
incubation period. Soils from all sites, rotations and water treatments were set up twice: To determine long-term recovery, one
set was subjected to only one stress cycle and was kept at 50% WHC for 6 weeks after the stress. The second set was subjected
to a total of 5 stress cycles.

To determine soil C and N pools and microbial enzyme activities in soils from Colorado and Maryland with the lowest and
highest MAP, respectively, we set up ten sets of each of the control, drought and flooding treatments for each of the sites. One
set from each site was destructively harvested before, at the peak of, right after, two weeks after, and 6 weeks after the first
and last stress cycle (Fig 1).

## 2.3 Soil C, water content, water holding capacity, pH

Samples for total C and N analysis were dried at 60°C for 24h and finely ground in a ball mill before subsamples were packed
in tin capsules and total C measured on an elemental analyzer (Costech Instruments ECS 4010) (Paul et al., 2001). Total C
content as well as water holding capacity was determined for all soils prior to the incubation experiment. Water holding
capacity was measured by determining soil water content after saturating the soils with water in a funnel with filter and letting
the excess water leach gravimetrically for two days while preventing evaporation by covering the funnels with parafilm (Paul
et al., 2001). Water con tent and pH were determined for all soils before the start of the incubation and during the 10 destructive
samplings for Colorado and Maryland soils. Water content was determined gravimetrically in sample aliquots that were dried
in a forced draft oven at 60°C for 24h. Soil pH was determined in a 1:5 soil to water mixture using a Mettler Toledo Seveneasy
pH Meter 20.





## 2.4 CO₂ and N₂O production

CO₂ production was measured daily for the first week and twice a week after that, as well as just before the destructive harvests.
For respiration measurements, cups were temporarily closed airtight with lids fitted with rubber septa that served as a sampling ports. We took 3 mL of the headspace using a syringe immediately after closing the sample and after 30 min to 2h, depending on the incubation duration within the experiment and the C content of the samples; longer incubation times were used at the end of the experiment and for soils with lower total C contents. The gas samples were immediately injected into an infrared gas analyzer (Li-cor LI 820) to measure $CO_2$ concentration. Rates of $CO_2$ production were calculated from the increase of $CO_2$
concentration in the headspace of the jar over time, accounting for jar and syringe volume and temperature, assuming linear increase between the two sampling time points. Cumulative respiration was calculated by using respiration rates measured at a certain time point and multiplying that flux with the number of days to the next respiration measurement and summing all resulting $CO_2$ emissions (Grandy and Robertson, 2007). To compare cumulative respiration among sites, values are expressed as µg $CO_2$-C per g soil C.

For the determination of N₂O production, which was measured at every destructive harvest, cups were inserted in pint-sized mason jars and sealed airtight with a lid fitted with a rubber septum. Right after closing the jars, a headspace sample of 30 mL was taken with a syringe and needle and transferred into pre-evacuated exetainers. The jars then remained closed for 24h before a second sample was taken and transferred to exetainers. N₂O concentration in the exetainers was determined using a Shimadzu GC-2014 equipped with an ECD detector. N₂O flux was calculated as the difference in N₂O concentration between
samples collected right after sealing and after 24h divided by the time of incubation and the amount of dry soil in the cup and accounting for jar and syringe volume and temperature.

## 2.5 Extractable organic carbon (EOC), total extractable N (TEN), ammonium, nitrate and microbial biomass carbon (MBC)

Extractable organic carbon and total extractable N was measured in 1M KCl extracts (15 mL) from approximately 2 g of soil
using a TOC-L CPH/CPN analyzer (Shimadzu). Ammonium and nitrate concentrations were measured in the same extracts by colorimetric assays as described by Hood-Nowotny et al. (Hood-Nowotny et al., 2010). Microbial biomass C was determined using chloroform-fumigation extraction (Brookes et al., 1985; Vance et al., 1987). Two g of fresh soil were fumigated in a desiccator under chloroform atmosphere for 24h in the dark and then extracted with 1M KCl. Extracts of fumigated samples were measured on the TOC-L CPH/CPN analyzer and microbial C was calculated as the difference in EOC
between the fumigated and the fresh soil extracts. Microbial C is presented without the use of a correction factor for extraction efficiency. EOC, TEN, NH₄, NO₃, and MBC were determined at every destructive harvest.

## 2.6 Enzyme activities

Potential extracellular enzyme activities were measured, with adaptations, as described in Schnecker et al. (2015). In short, 2g of soil were suspended and homogenized in 100mL 100 mM sodium acetate buffer at pH 5.5. For each sample and each



enzyme, 5 wells of a black microtiter plate were filled with 200 μL of the soil slurry. The respective wells were amended with MUF (4-methylumbelliferyl) labeled substrates: β-D-glucopyranoside for β-glucosidase (BG), β-D-cellobioside for cellobiohydrolase (CBH) and N-acetyl-β-D-glucosaminide for N-acetyl-glucosaminidase (NAG). L-Leucine-7-amido-4-methyl coumarin was used as substrate for leucine-amino-peptidase (LAP). Plates for the assays of BG, CBH, NAG, and LAP were incubated for 140 min. Afterwards, activity was measured fluorimetrically (excitation 365 nm and emission 450 nm). Phenoloxidase (POX) and peroxidase (PEX) activities were measured using L-3,4-dihydroxyphenylalanine (DOPA) as substrate in a photometric assay. Three times 1 mL of the original soil slurry was mixed with 1 mL of a 20 mM DOPA solution. After shaking and centrifuging, two wells of each transparent microtiter plate were filled with 250 μL of the supernatant. One of these wells additionally received 10 μL $H_2O_2$ (0.3%) for determination of peroxidase activity. Plates for oxidative enzyme activities were measured photometrically (absorbance 450 nm) at the beginning and after incubation for 20 hours. PEX activity was calculated as the difference in the increase in color during the incubation time between the wells with and without $H_2O_2$ addition. All other enzyme activities were calculated as the increase in color or fluorescence during the incubation time. Potential enzyme activities were determined at every destructive harvest.

## 2.7 Labile Carbon

To quantify labile soil C we used the permanganate oxidizable C (POXC) method (Weil et al., 2003) as described in Culman et al. (2012). In short, 2.5 g of air-dried soil were mixed with 18 mL of deionized water and 2 mL of 0.2 M $KMnO_4$ stock solution and shaken for 2 min at 240 oscillations per minute on an oscillating shaker. Tubes were removed from the shaker and allowed to settle for 10 min. After 10 min, 0.5 mL of the supernatant were mixed with 49.5 mL of deionized water. An aliquot (200 μL) of each sample was loaded into a 96-well plate containing a set of internal standards, a soil standard and a solution standard (laboratory reference samples). Sample absorbance was read with a spectrometer at 550 nm. POXC was determined at every destructive harvest for Colorado and Maryland soils.

## 2.8 Statistics

To evaluate the effect of the specific stress treatments, we calculated response ratios of all variables measured during destructive harvests as the values for the treated samples divided by the values for the control samples under constant water conditions. To evaluate differences between control and treatment samples, we performed two-sample comparison tests (t-test, Welch-test, or Mann-Whitney-U-test as appropriate for each variable's normality and homogeneity of variance). We further used all data measured at destructive harvests, with the exception of water content, individually for each harvest date and including all water treatments at peak stress and right after the stress during the first and last stress cycles (days 6, 14, 120, and 127) in Principal Components Analysis (PCA). We used one-way ANOVA and Tukey HSD as a post-hoc test on the first two axes of the PCAs to evaluate differences among water treatments, crop rotation complexity treatments, and their interaction. Before analysis, data were log-transformed or rank-normalized to meet the assumptions for ANOVA. Differences and correlations were assumed to be significant at $p < 0.05$. Statistics were performed in R 3.3.2 (R Development Core Team, 2013).



## 3 Results

### 3.1 Differences among sites and rotations

The sites used in this study represent a gradient in MAP from an arid system in Colorado (MAP 421mm) to a site with relatively

high MAP in Maryland (1192 mm), with South Dakota (580 mm) and Michigan (892 mm) providing intermediate MAP. Soil organic carbon (SOC) content varied greatly between sites, being highest in South Dakota, followed by Maryland, Colorado, and Michigan (Table 1). Significant differences in SOC content between rotation lengths could only be found in Maryland where soils from the complex rotation had an average of 1.3% OC and soils from the simple rotation length had 1.0% OC; notably, this was also the only site that included a perennial crop in the complex rotation.

### 3.2 Heterotrophic respiration response to drought and flooding

Respiration decreased significantly in response to drought in soils from all sites and crop rotation complexities, and returned to control levels as soon as microcosms were returned to 50% WHC after the first (day 1-15) and fifth (day 113-127) stress cycle (Fig. 2). In some cases (Maryland first and fifth stress, South Dakota fifth stress and Michigan fifth stress) respiration in re-wetted microcosms exceeded respiration in control microcosms. Soils from simple and complex rotations did not differ in

the response to drought, but in some cases differed in their recovery from the stress, most notably in soils from Maryland and South Dakota where soils from simple rotations showed lower respiration rates. In general soils experienced slight, but mostly not significant increases in respiration in response to flooding.  In the Colorado simple rotation, the Maryland simple rotation, the Michigan complex rotation and both South Dakota rotations, flooding significantly increased respiration only on day 4 and thereafter was indistinguishable from the control until the end of the first stress. Soils from the complex rotation in South

Dakota further showed an increase in respiration during the recovery from flooding after the fifth stress.

### 3.3 Cumulative respiration

We measured cumulative respiration to estimate soil carbon loss. Respiration was highest in soils from Colorado, followed by those from Maryland, Michigan, and South Dakota (Fig. 3). During the first stress cycle, soils from all sites lost significantly less $CO_2$ under drought compared to control and flooding treatments (Fig. 3b), but $CO_2$ did not differ between flooded and

control soils at any site for a given rotation treatment. During the fifth stress cycle $CO_2$ losses were significantly lower with drought compared to control and flooding in soils from Colorado and South Dakota under both rotation regimes, and drought-stressed soils from the Maryland complex rotation lost significantly less $CO_2$ than flooded microcosms. Total loss of C as $CO_2$ (calculated per g SOC) over the entire incubation period tended to be lower in microcosms experiencing repeated drought compared to control and flooded microcosms (Fig. 3a). However, drought-stressed soils under complex rotation in Colorado

lost significantly less C as $CO_2$ than control and flooded soils, and soils from simple rotations in South Dakota lost more C when flooded compared to drought, while control soils were not significantly different from either stress treatment.





### 3.4 Effect of drought and flooding on soil C and N pools and microbial functions in the driest (CO) and wettest (MD) locations

Flooding and drought caused significant changes in soil N and C pools, microbial enzyme activities, and nitrous oxide
production. Except for experimentally-manipulated water content and a decrease in respiration during drought, no variables
changed consistently and significantly among soils from Colorado and Maryland (Fig. 4 and 5). In general, drought tended to
decrease measured parameters, while flooding increased them.  In Colorado, all enzyme activities decreased with drought
while NAG alone increased with flooding. This was not the case in Maryland soils, where enzyme activities remained constant
through water stress or changed only after the stress was over. Only a few parameters changed similarly during the first and
fifth stress periods. LAP decreased during the first and fifth drought stresses in the simple Colorado rotation treatment.
Production of $N_2O$ increased in response to the first and fifth flooding in the Maryland complex rotation soil; in this rotation
treatment NAG decreased after both the first and fifth drought. All other parameters affected by the stress treatment changed
only during either the first or fifth stress in single site-rotation combinations at peak stress or following the stress.

### 3.5 Recovery from stress

Samples grouped by flooding and drought treatments in measured response parameter ordination space (PCAs; Fig. 6).
Samples clustered similarly, and treatment differences on individual PCA axes were significant (Table 2), during the first (Fig.
6) and fifth stress (Fig. 7). In both cases these differences were no longer significant after the end of drought and flooding,
when microcosms returned to 50% WHC (Fig 6 and 7, Table 2). In the case of Maryland samples, significant differences could
be found between simple and complex rotations during and after the first and fifth stress periods. Differences between rotations
in Colorado were only found after the fifth stress period.

### 4 Discussion

Drought and flooding represent severe stressors for soil microbial communities. In our study of soils under low and high
diversity crop rotation regimes from agricultural sites across the U.S., we found that short-term drought—and to a lesser degree,
flooding—led to overall significant and stress-specific changes in microbial processes and functions. Respiration was strongly
reduced in all sites and rotation treatments during drought stress. Flooding caused $N_2O$ production in soils from three of four
sites during the first flooding event, though this effect remained after the fifth flooding only in soils from Maryland. We found
potential site-dependent legacy effects for $CO_2$ release at the drier Colorado site where we found the highest specific
cumulative respiration rates of all sites (Fig. 3) and a consistent lack of the Birch effect (Birch, 1958).  Interestingly, soils
managed under rotations of only two crops versus 3-4 crops did not significantly differ in their response to stress. In general,
all soils—irrespective of site and rotational complexity—responded strongly to drought and flooding but recovered quickly to
control levels when water content returned to 50% of water holding capacity, suggesting that the investigated agricultural soils
microbial communities are highly resilient to water stress.



## 4.1 Response to water stress

All soils had significantly decreased respiration rates in response to drought during the first stress cycle with the strongest
relative decreases at the Colorado site. Compared to drought, flooding had a smaller effect on microbial respiration. In general
flooding slightly increased respiration rates, but this was only significant during the first stress cycle for one to two days and
was not consistent across rotation treatments. Flooding may increase respiration rates in part due to increased connectivity and
availability of previously untapped DOC sources to microorganisms (Schimel, 2018 and therein). Cumulative C losses were
significantly different among sites. Surprisingly, despite having the highest SOC contents, soils from South Dakota lost the
least C as $CO_2$ over the whole incubation period. Differences in cumulative respiration between sites could be related to
differences in minerology and soil texture (Saidy et al., 2015; Schmidt et al., 2011), microbial community composition (Babin
et al., 2013), or chemistry of plant and fertilizer inputs (McDaniel and Grandy, 2016).

Aside from $CO_2$ production during drought no factor investigated in the two soils from climate extremes (Colorado and
Maryland) changed consistently in response to drought and flooding (Fig. 4). In Colorado soils of both rotation lengths, LAP
decreased with drought, which was not the case in soils from Maryland. All flooded Colorado soils produced $N_2O$, while this
was only the case in the complex rotation in Maryland. Beside these site-specific effects, we could not find drought or flooding
effects that occurred in either rotation treatment at both sites. Extracellular enzyme activities in particular remained relatively
unaffected by the applied stresses. A reason for this might be the stabilization of enzymes on soil minerals, which might protect
them against drought and flooding (Allison and Jastrow, 2006; Kramer et al., 2013). This stabilization might also explain why
our findings are in contrast to results from temperature stress experiments in plant litter (Mooshammer et al., 2017).

However, when soil samples were ordinated in response parameter space in a PCA, a clear stress treatment effect emerged in
Colorado and Maryland soils irrespective of the rotation treatment (Fig. 4 and Fig. 5, Table 2). Maryland but not Colorado
soils showed an additional rotation effect: samples from simple and complex rotations at this site separated in the PCA, both
during and after drought and flooding events. This was most likely related to higher SOC content and co-varying soil properties
in the complex crop rotation soils, which we found only in Maryland. The experimental field in Maryland had the most complex
crop rotation (four crops) and was the only site where the complex rotation included a perennial crop and fertilization with
poultry litter, both are effective methods to increase soil C stocks and soil health (Ashworth et al., 2018; King and Blesh, 2018)
and might even be more effective than increasing cropping diversity alone (McDaniel et al., 2014).

One reason for the apparently mild reaction to drought and flooding in this experiment might be the duration of the stress we
applied. Stress slowly applied over several days and lasting less than two weeks in total might be similar to conditions that
microorganisms in the investigated soils experience frequently in the field. Stress effects have indeed been found to strongly
vary with the duration of stress as well as its intensity (Tiemann and Billings, 2011, 2012).



## 4.2 Recovery from stress

In our experiment, only soils from Maryland showed a stress-induced increase in respiration during recovery, compared to
unstressed control soils; specifically, respiration increased in soils from the complex Maryland rotation in the first day
following stress and in the simple rotation after three days. The lack of a Birch effect in all other soils might be because we re-
wetted the soil in small increments over the course of three days rather than a flush rewetting, which often results in a large
increase in respiration (Birch, 1958; Göransson et al., 2013). It might also be the case that the duration of desiccation was too
short to lead to a pronounced $CO_2$ pulse at re-wetting (Unger et al., 2010).
Like respiration rates, most other measured parameters returned to control levels following stress; this was also apparent in the
PCA where dried and flooded soils were indistinguishable from control soils at a constant water content of 50% WHC. This
indicates that, while all soils were significantly affected by the stress treatment, they are highly resilient and recovered quickly
from stress. This is in accordance with Kaurin et al. (2018) who found that microbial communities in agricultural soils
recovered after rewetting even after severe and prolonged drought periods and Barnard et al. (2013) who found similarly quick
recovery of the soil microbial community after rewetting of dry grassland soils.

## 4.3 Adaptation to stress

As during the first stress period, respiration rates declined in soils from all sites and rotations during the fifth drought cycle. In
contrast, respiration was unaffected by flooding, with the exception of a slight increase in the Michigan complex rotation 5
days after flooding. After the end of the fifth drought cycle, in contrast to the first cycle, we found clear increases in respiration
when soils from Maryland, South Dakota, and Michigan returned to 50% WHC. Both observations might be related to the
absence of plant inputs in our incubation experiment. Plants have been shown to strongly respond to drought (Fuchslueger et
al., 2014; Kaisermann et al., 2017), but will provide some amount of C even if rhizodeposition is reduced under drought
(Canarini and Dijkstra, 2015). During the vegetation period, such a continuous supply of root exudates might prevent an
increase in respiration at the end of a stress event when connectivity between microbes and substrate is re-established, such as
that which we observed after the fifth but not the first stress. Management practices that extend the vegetation period and
minimize fallow periods might help maintain a constant supply of DOC to soil microorganisms and thereby buffer their
response to drought and flooding.

We also found site-specific differences between the first and the fifth stress cycles. In soils from Colorado from both rotation
regimes, NAG was significantly reduced in the drought treatment compared to the control. In contrast, drought reduced total
extractable N in Maryland soils while flooding released $N_2O$ in both Maryland rotations. In Colorado soils no $N_2O$ production
could be detected by the fifth stress cycle. This might have been caused by a depletion of the dissolved substrate for $N_2O$
production, or could be related to plant-induced differences in microbial community composition (Hammerl et al., 2019) that
had faded after 160 days of incubation.



The lack of microbial adaptation to the re-occurring stress in all except the Colorado soils might be interpreted as an already
existing adaptation to conditions mimicked in our experiment or could again be ascribed to the modest stress events of our lab
experiment compared to the larger environmental fluctuations these soils experienced in situ.

## 4.4 Summary

In this study we found that drought—and, to a lesser extent, short-term flooding—significantly affected respiration rates at all
sites, and at some sites had additional effects on some microbial enzyme activities, soil C and N pools, and nitrous oxide
emissions. Furthermore, an increase in crop rotational diversity did not lead to generally different responses of soils to short-
term drought and flooding, even at the one site where increased crop rotation complexity also caused an increase in SOC.
While soil function reacted significantly during stress events, all soils recovered quickly and returned to control levels once
the stress ended. This indicates that soil microbial processes in these agricultural soils, collected from variable climate regions
within the United States, are highly resilient to short term drought and flooding. Future experiments should also include plants
as they are an important component of agroecosystems in the field and could strongly influence DOC and DON availability.
Our laboratory study focused on the soil-microbe system and showed that, at least in the absence of plants, microbial functions
and activities are highly resilient to drought and flooding and recover quickly from stress.

## 5 Acknowledgements

This research used samples and data from the Long-Term Agroecosystem Research (LTAR) network, which is supported by
the United States Department of Agriculture. The Michigan and Maryland soils used in the study are from LTAR sites. LTAR
is supported by the United States Department of Agriculture. Long-term crop rotation treatments at the South Dakota site are
managed and maintained by Dr. Shannon Osborne, USDA-ARS. This research was funded by the National Institute of Food
and Agriculture, U.S. Department of Agriculture, under award number 2014-67019-21716.

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

**Figures and Tables:**

**Table 1:** Information on the sites used in the laboratory incubation experiment. MAT is mean annual temperature, MAP is mean annual precipitation, SOC is soil organic C content. Asterisks indicate significant difference between simple and complex rotations.


| Site | Coordinates | MAT (°C) | MAP (mm) | Soil texture | Plants in rotation | | SOC % | |
|------|-------------|----------|----------|--------------|--------------------|--------------------|-------|-------|
| | | | | | Complex rotation | Simple rotation | Complex rotation | Simple rotation |



| USDA-ARS Akon, Colorado (CO) | 40°07'40"N 103°07'58"W | 9.8 | 421 | weld silt loam | wheat-corn-millet-pea | wheat-millet | 0.7 | 0.8 | |
| USDA-ARS Brookings, South Dakota (SD) | 44°20'27"N 96°47'18"W | 6.2 | 580 | sandy clay loam | corn-soybean-wheat-sunflower | corn-soybean | 2.2 | 2.1 | 505 |
| W.K. Kellogg Biological Station, Michigan (MI) | 42°24'23"N 85°22'32"W | 8.9 | 893 | loam and sandy loam | corn-soybean-wheat | corn-soybean | 0.8 | 0.8 | 510 |
| USDA-ARS Beltsville, Maryland (MD) | 39°01'27"N 76°55'29"W | 13.6 | 1192 | silt loam | corn-soybean-wheat-alfalfa-alfalfa-alfalfa | corn-soybean | 1.3* | 1.0 | 515 |


**Table 2:** Results from Analysis of Variance of treatments on Axis from PCAs as seen in Figure 3 and 4. Bold values represent significant differences between treatments (drought, flooding and control).

| | PC1 | | | | | | PC2 | | | | | |
| | treatment | | rotation | | interaction | | treatment | | rotation | | interaction | |
| | *F* | *p* | *F* | *p* | *F* | *p* | *F* | *p* | *F* | *p* | *F* | *p* |
|---|---|---|---|---|---|---|---|---|---|---|---|---|
| Colorado day 6 peak stress | **35.37** | **<0.001** | 0.001 | 0.973 | 0.370 | 0.700 | 0.262 | 0.775 | 1.560 | 0.240 | 0.303 | 0.745 |
| Colorado day 14 recovery | 0.047 | 0.954 | 2.054 | 0.190 | 0.055 | 0.947 | 1.296 | 0.325 | 4.627 | 0.064 | 0.461 | 0.646 |
| Colorado day 120 peak stress | 3.214 | 0.076 | 0.431 | 0.524 | 0.109 | 0.898 | **4.124** | **0.043** | **5.291** | **0.040** | 0.813 | 0.466 |
| Colorado day 127 recovery | 0.795 | 0.476 | 2.107 | 0.175 | 0.017 | 0.983 | 0.496 | 0.622 | 4.416 | 0.060 | 0.553 | 0.591 |



| | | | | | | | | | | | | |
|---|---|---|---|---|---|---|---|---|---|---|---|---|
| Maryland day 6 peak stress | 0.476 | 0.631 | **98.36** | **<0.001** | 1.152 | 0.344 | **21.980** | **<0.001** | 0.939 | 0.349 | 1.765 | 0.207 |
| Maryland day 14 recovery | 0.024 | 0.977 | **46.56** | **<0.001** | 0.192 | 0.827 | 2.068 | 0.159 | 0.400 | 0.536 | 0.814 | 0.461 |
| Maryland day 120 peak stress | **6.872** | **0.007** | **47.63** | **<0.001** | 0.979 | 0.397 | **80.916** | **<0.001** | **22.038** | **<0.001** | 1.945 | 0.175 |
| Maryland day 121 recovery | 0.622 | 0.548 | **89.13** | **<0.001** | 2.043 | 0.159 | 0.583 | 0.568 | 0.025 | 0.876 | 0.470 | 0.632 |


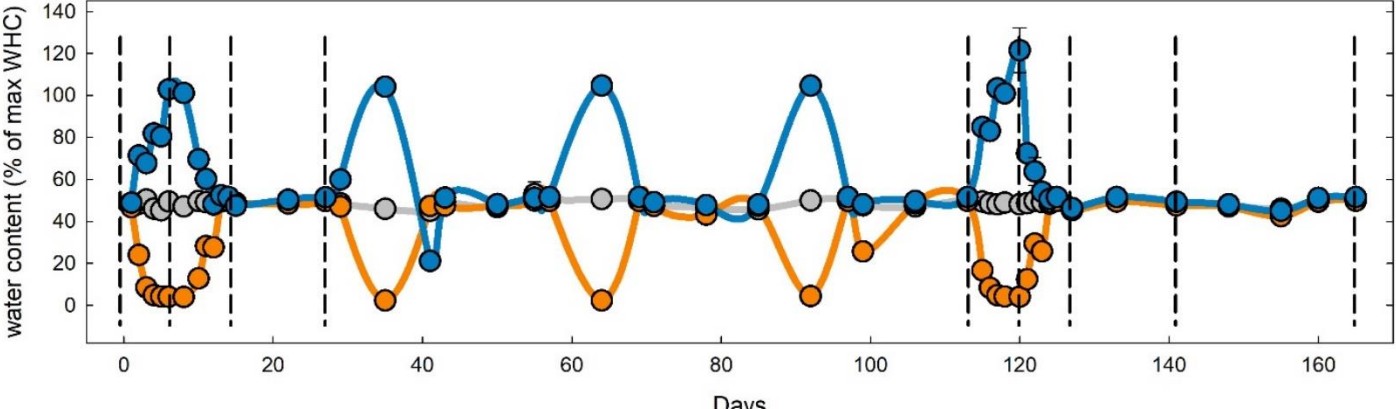

**Figure 1** Mean water holding capacity of all four sites and rotation lengths during the course of the experiment. Symbols and lines in blue represent flooding treatment; orange, drought treatment; and gray, control. Dashed vertical lines represent destructive harvests (1-4 and 6-10) of subsets on days 6, 14, 27, 113, 120, 127, 141, and 165. To study long-term recovery, an additional set of samples (not shown)
underwent only the first stress cycle and was subsequently kept at 50% WHC until harvest on day 55 (harvest 5).



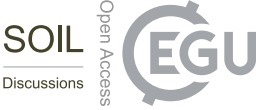

**Figure 2** Soil respiration rates during the first (left panels) and fifth (right panels) stress cycles relative to the control at 50% WHC. Blue symbols represent microcosms exposed to flooding; orange symbols represent drought treatment. Open symbols are simple rotations and
filled symbols are the complex rotations at the respective sites. * indicates that the treatment significantly changed respiration with respect to the control. *C and *S mean that only soil samples from complex or simple rotations, respectively, had significantly different respiration rates than the respective control. If indicators for significance are above the graphs, they refer to the flooding treatment, when below they refer to the drought treatment. Significant difference was assumed at $p < 0.05$.







**Figure 3** Cumulative respiration calculated per g SOC over (a) the whole experimental period (b) during the first stress period (c) during the 5th stress period. Statistically significant differences among control, drought and flooding treatments for a given site and rotation complexity (C is complex rotation, S is simple rotation) are indicated by letters. Capital letters indicate differences among control treatments of all sites and rotations.

| | Harvest 2 (day 6) | | | | Harvest 3 (day 14) | | | | Harvest 4 (day 27) | | | | Harvest 5 (day 55) | | | |
|---|---|---|---|---|---|---|---|---|---|---|---|---|---|---|---|---|
| Site | Colorado | | Maryland | | Colorado | | Maryland | | Colorado | | Maryland | | Colorado | | Maryland | |
| rotation length | C | S | C | S | C | S | C | S | C | S | C | S | C | S | C | S |
| **response to drought** | | | | | | | | | | | | | | | | |
| water content | 0.10 | 0.09 | 0.04 | 0.15 | | | | | | | | | | | | |
| pH | | | | | | | | | | | | | | | | |
| cellobiohydrolase | | | | | | | 0.84 | | | | | | | | | |
| beta-glucosidase | 0.72 | | | | | | | | 1.28 | | | | | | | |
| N-acetyl-glucosaminidase | | | | | | | 0.65 | | | | | 0.75 | 0.90 | | | |
| leucine-aminopeptidase | 0.80 | 0.81 | | | | | | | | | | | | | | |
| phenol-oxidase | | 1.14 | | | | | | | | 1.12 | | | | | | |
| CO₂ | 0.07 | 0.05 | 0.03 | 0.02 | | | | | | | | | | | | |
| N₂O | | | | | | | | | | | | | | | | |
| Nitrate | | | 0.76 | | | | 0.68 | | | | | | 0.95 | | | |
| Ammonium | | | 2.06 | | | | | | | | | | | | 0.79 | |
| extractable organic C | | | | | | | | | | | | | | | | |
| total extractable N | | | | | | | | | | | | | | | 0.66 | |
| microbial biomass C | | | | | | | | | 0.97 | | | | | | | |
| POX C | | | | | | | | | | | | | | | | 0.75 |
| | | | | | | | | | | | | | | | | |
| **response to flooding** | | | | | | | | | | | | | | | | |
| water content | 1.70 | 1.66 | 1.61 | 4.88 | | | | | 1.01 | | | | | | | |
| pH | | 1.04 | | | | | | | | | | | | | | |
| cellobiohydrolase | | | | | | | | | | | | | | | | |
| beta-glucosidase | | | | | | | | | | | | | | | | |
| N-acetyl-glucosaminidase | | | | | | | | | | | | | | | | |
| leucine-aminopeptidase | 1.16 | | 1.25 | | | | | | | | | | | | | |
| phenol-oxidase | | 1.12 | | | | | 1.06 | | | | | | | | | |
| CO₂ | | | | | 0.88 | | | | | | | | | | | |
| N₂O | 112954 | 122014 | 916 | | | | | | | | | | | | | |
| Nitrate | | | | | | | 1.75 | | 0.70 | | | | | | | |
| Ammonium | | | | | | | | | | | | | | | | |
| extractable organic C | | | | | | | | | | | | | 0.80 | | | |
| total extractable N | | 0.90 | | | | | 4.64 | | | | | | | | | |
| microbial biomass C | | | 2.05 | | | | | | 1.33 | | | | | | | |
| POX C | 1.19 | | | | | | | | | | 1.13 | | | | | |


**Figure 4** Mean response ratios of soil C and N pools, microbial enzyme activities and CO₂ and N₂O production in response to drought and flooding, during (day 6), immediately after (day 14), 2 weeks (day 27) after and 6 weeks (day 55) after the first stress cycle. orange colors indicate a reduction relative to the control, green colors represent an increase. Only significant differences (p<0.05) are shown. Response ratios are calculated as the value for a given treatment divided by the value of the respective control at 50% WHC. C is complex rotation, S is simple rotation, POXC is permanganate oxidizable C.





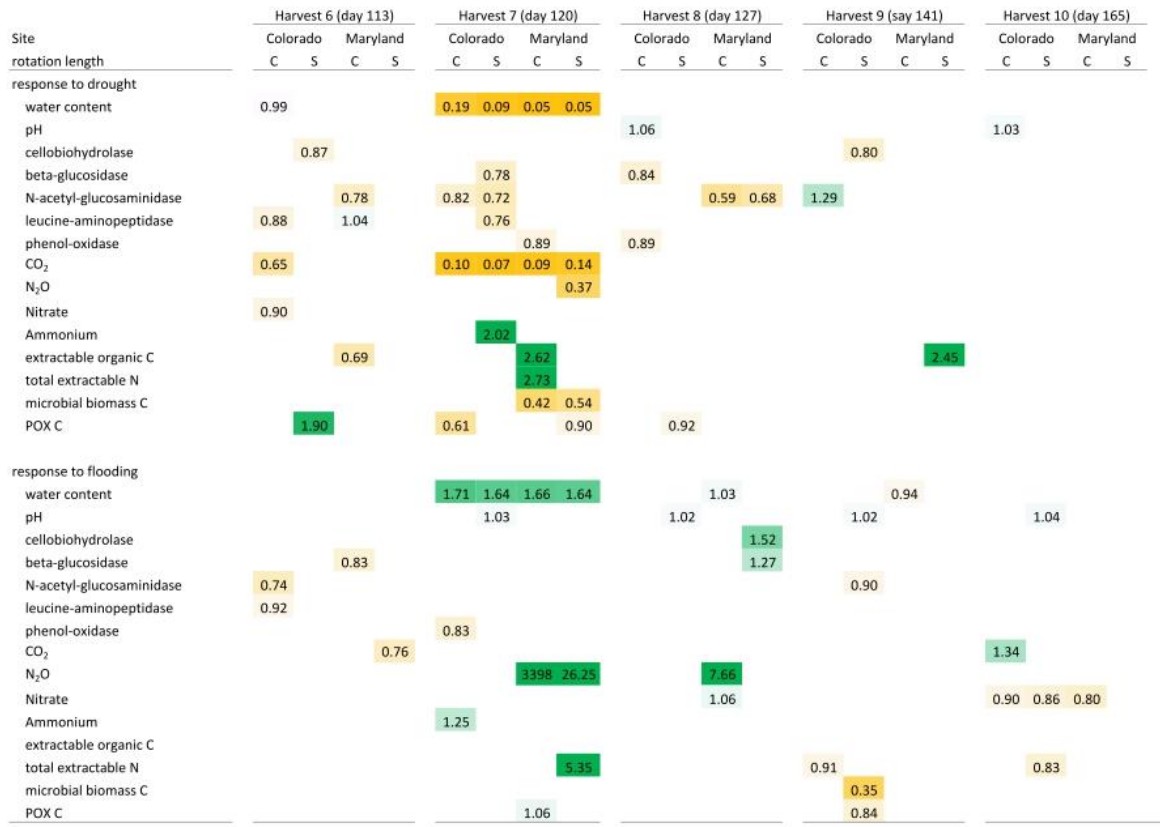

**Figure 5** Mean response ratios of soil C and N pools, microbial enzyme activities and $CO_2$ and $N_2O$ production in response to drought and flooding, before (day 113), during (day 120), immediately after (day 127), 2 weeks (day 141) after and 6 weeks (day 165) after the fifth stress cycle. orange colors indicate a reduction relative to the control, green colors represent an increase. Only significant differences ($p<0.05$) are shown. Response ratios are calculated as the value for a given treatment divided by the value of the respective control at 50%WHC. C is complex rotation, S is simple rotation, POXC is permanganate oxidizable C.





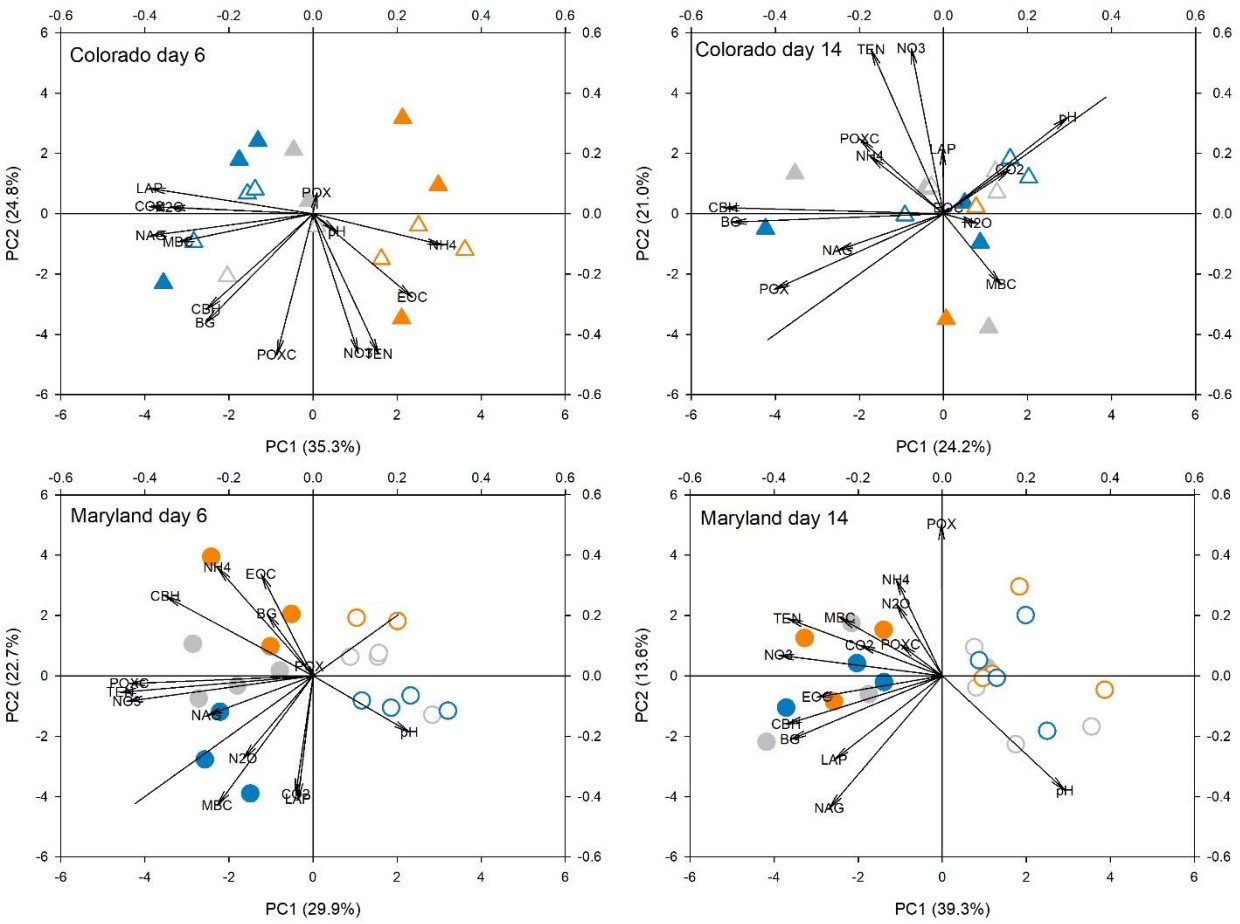

**Figure 6** Principal components analysis of all response parameters at first peak stress (day 6) and following the first stress (day 14) in soils from Colorado and Maryland. Blue symbols are flooding treatment, orange symbols are drought treatment. Gray symbols are control. Open symbols represent simple and filled symbols represent complex rotations. Significant differences between treatments along the axes are shown in table 2. Included parameters are pH, extractable organic carbon (EOC), total extractable nitrogen (TEN), microbial biomass C (MBC), $NH_4$, $NO_3$, activities of β-glucosidase (BG), cellobiohydrolase (CBH), N-acetyl-glucosaminidase (NAG), leucine-amino-peptidase (LAP, phenoloxidase (POX) and peroxidase (PEX), respiration ($CO_2$), N$_2$O production ($N_2O$) and permanganate oxidizable C (POXC)



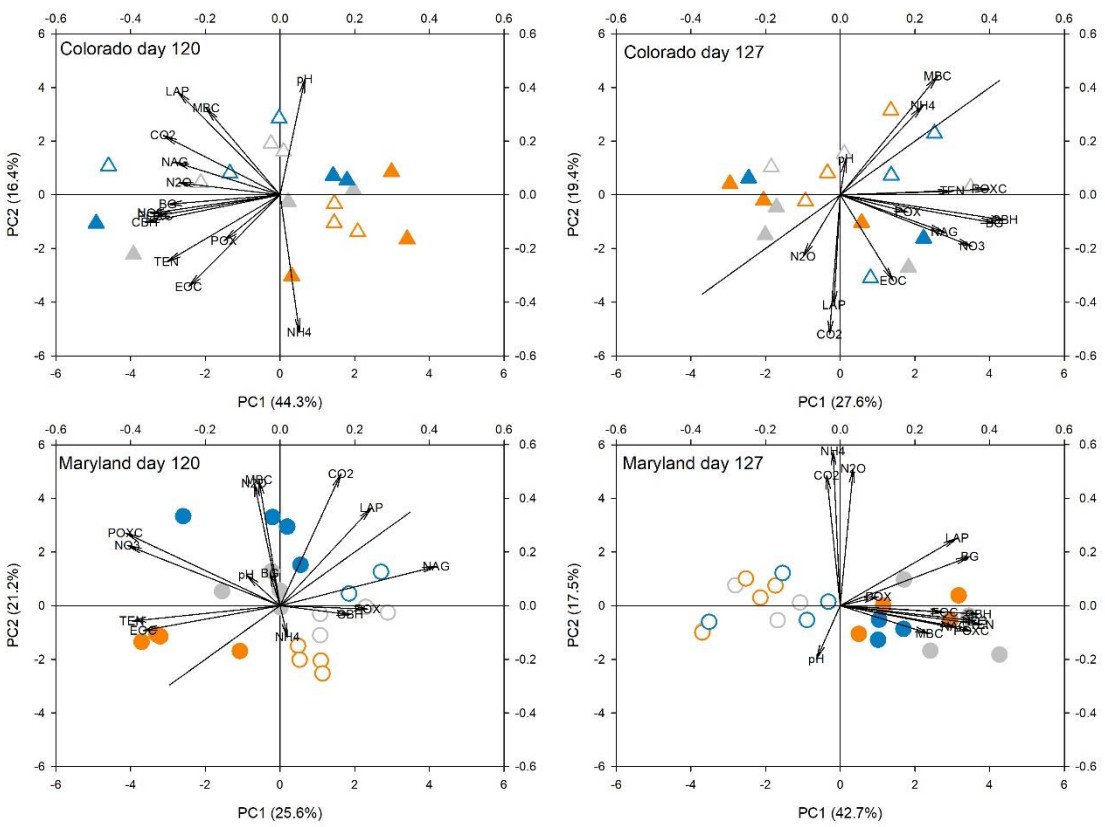

**Figure 7** Principal components analysis of all response parameters at fifth peak stress (day 120) and following the fifth stress (day 127) in soils from Colorado and Maryland. Blue symbols are flooding treatment, orange symbols are drought treatment. Gray symbols are control. Open symbols represent simple and filled symbols represent complex rotations. Significant differences between treatments along the axes are shown in table 2. Included parameters are pH, extractable organic carbon (EOC), total extractable nitrogen (TEN), microbial biomass C (MBC), NH4, NO3, activities of β-glucosidase (BG), cellobiohydrolase (CBH), N-acetyl-glucosaminidase (NAG), leucine-amino-peptidase (LAP, phenoloxidase (POX) and peroxidase (PEX), respiration ($CO_2$), N2O production (N2O) and permanganate oxidizable C (POXC).

575