# Peer review of "Microbial activity responses to water stress in agricultural soils from simple and complex crop rotations"

_SOIL, 2021_

## Author Response (AR1)

Thank you for your comments. We think that they will improve the manuscript. Below we have replied to your specific comments and have adapted the manuscript accordingly.

Reviewer I

Dear Schnecker et al,

I have reviewed your manuscript "Microbial activity responses to water stress in agricultural soils from simple and complex crop rotations" submitted to SOIL. The study examines soil respiration in a lab incubation of soils collected under simple row-crop rotations and complex/multi-species rotations. Soils spanned a moisture gradient from Colorado to Maryland. The incubations replicated drought and flooding stress events. Additional response variables: microbial biomass, enzyme activity, and $N_2O$ production were collected from the driest and wettest sites. Responses were consistent to stress events and generally returned to control levels after stress was removed. Cropping rotation did not affect any of the soil or microbial responses. Given the variation in agricultural management across these sites, which reflects variation in practices across this geographic scale, the data area analyzed appropriately, and results interpreted reasonably.

It might strengthen the paper to consider the field sites as spanning both a climatic gradient and soil texture gradient. Soil organic carbon numbers match the soil texture gradient more strongly than the precipitation gradient (e.g. sandy clay loam in SD had the highest SOC and sandy loam in MI had the lowest), which in interesting in an of itself, as soil physical properties seem to be a stronger driver than plant inputs and climate. Since the responses here are from soil incubations without plants, it might be interesting to explore the effects of soil texture more deeply.

A few additional comments:

- If you have access to measured texture (%sand), it would be informative to add that to table 1 to better understand the underlying soil physical gradient.

  >> We agree that physical soil properties are playing an important role in C dynamics and most likely also water dynamics and thus the response to changes in water content. It is interesting that OC is highest in SD with the nominally heaviest soils and lowest in MI with the nominally lightest one. The other two sites however don't fit as nicely with both having silty loam and similar dynamics in terms of their water contents during drought and flooding (Table S1) but while CO has as low OC as SD, MD soils are significantly higher. We also only found significant changes in OC through management in MD, which has an intermediate soil texture. Soils from these two sites also responded differently to the water stress. Also, C loss through respiration calculated per g SOC was higher in CO than at the other sites, which all showed similar respiration rates. A pattern which seems not to reflect a soil texture gradient. Unfortunately, we do not have data on the specific soil textures from the soil samples we used for incubation but we think, based on the comparison of CO and MD soils and the results on respiration, that a texture gradient alone might not explain the differences between sites we found. Maybe detailed investigations of soil physical properties, including texture, minerology and soil pore

properties might help explain some of the results we got but these analyses should be carried out on a larger gradient and are probably topic of a separate study. We have however added some thoughts on soil physics in the discussion.  Lines 268-271

- How long have the cropping rotations been in effect before sampling? It would be helpful to have that information in the methods so the reader doesn't have to track it down in the referenced studies.

  >> We have added the year, when the field treatments were established to table 1

- The supplemental information includes Table S2. Is there a Table S1?

  >> Thank you for noting this, we have now renamed the table to S1

Reviewer II

This is a well-designed and well-executed experiment to separately examine the effects of drying and flooding on microbial activity in soils from across a moisture gradient and under simple or complex crop rotation strategies. The aims and approach are satisfactorily presented, and the analyses and interpretations are appropriate. I have a few suggestions to improve the paper.

**Specific comments**:

- Since the abstract includes microbial biomass among the key parameters analyzed, I would like to see it addressed explicitly in the results section. As it stands, this variable is lumped with "other parameters" in section 3.4. Though included in Figures 4 and 5, the interpretation is challenging for the reader.

  >> Thank you for the comment. We have added information on MBC to the results section. Lines 227-232

- In the experimental setup, section 2.2, I found it hard to keep track of "sets" to determine the degree of replication within a sampling site and treatment. Please add clarifying details.

  >> We added some sentences for clarification in the methods section. Lines 100, 113-115

**Technical corrections**:

- Line 55: Remove "of" before nitrous oxide. >> Done. Line 55
- Line 72: Change "crops yields" to "crop yields." >> Done. Line 72
- Line 77: Change "drought and flooding" to "drought or flooding." >> We would like to keep the wording here as it is, since the point we wanted to make is that in most drought experiments, rewetting causes flooding, which we explicitly tried to avoid in our experiment.

- Line 80: Be consistent with line 94, where simple means two crops in rotation, not one or two. >> Done. Line 80
- Line 83: Change "drought or flooding" to "drought and/or flooding." >> Done. Line 83
- Line 99: Adjust format on degrees C. >> Done. Line 99
- Line 121: Change "was determined" to "were determined." >> Done. Line 123
- Line 131: Here and elsewhere, be consistent in spacing between numbers and units. >> Done. Lines 121, 128, 133, 144, 147, 155, 161, 197
- Line 149: Change "was measured" to "were measured." >> Done. Line 151
- Line 151: Remove authors names from parenthetical citation. >> Done. Line 153
- Line 168: Here and elsewhere, use subscripted numbers in chemical formulas. >> Done Lines 170,172, 177
- Line 251: Make use of your abbreviation, WHC. >> Done. Line 257
- Line 277: Use a semicolon to prevent a run-on sentence. >> We changed the sentences here. Line 289